# ENHANCING GROUP FAIRNESS IN FEDERATED LEARNING THROUGH PERSONALIZATION

## ABSTRACT

Instead of producing a single global model for all participating clients, personalized Federated Learning (FL) algorithms aim to collaboratively train customized models for each client, enhancing their local accuracy. For example, clients could be clustered into different groups in which their models are similar, or clients could tune the global model locally to achieve better local accuracy. In this paper, we investigate the impact of personalization techniques in the FL paradigm on local (group) fairness of the learned models, and show that personalization techniques can also lead to improved fairness. We establish this effect through numerical experiments comparing two types of personalized FL algorithms against the baseline FedAvg algorithm and a baseline fair FL algorithm, and elaborate on the reasons behind improved fairness using personalized FL methods. We further provide analytical support under certain conditions.

## 1 INTRODUCTION

In recent years, Federated Learning (FL) has emerged as a pivotal paradigm with the objective of collaboratively training a global model while safeguarding data privacy by circumventing direct data access from local clients (Kairouz et al., 2021). FL offers the distinct advantage of potentially yielding superior model performance compared to models trained solely at local levels, owing to its capacity to aggregate knowledge from diverse clients. One of the most popular FL algorithms, FedAvg, introduced by McMahan et al. (2017), demonstrates commendable performance, particularly in scenarios where samples from different clients are independently and identically distributed (IID). However, real-world applications often involve heterogeneous data points across clients, a situation where the performance of FedAvg can considerably deteriorate (Li et al., 2019b). Moreover, the classical FL approach yields a single global model, devoid of customization for individual clients. Consequently, clients with heterogeneous datasets may encounter local accuracy degradation. To address these limitations, a spectrum of personalized techniques has been proposed in the FL literature, as discussed further in Section 2. These techniques are designed to enhance the local accuracy of the learned models, while keeping some of the benefits of collaborative learning.

In addition to handling issues of data heterogeneity, training FL models that can uphold societal values, as formalized through notions of *algorithmic fairness* (Barocas et al., 2019), is an important consideration. Consider as an example the scenario where FL is used to train a foundation model/LLM on local datasets, contributed by diverse participants from different regions and communities. Without careful oversight, the resulting model will favor language and content preferred by the majority contributors, often disregarding the unique linguistic nuances and cultural contexts of minority groups (Durmus et al., 2023). As another example, existing research (Kirdemir et al., 2021) finds structural and systemic bias in YouTube video recommendation systems. In this case, the collaboratively trained recommendation model begins to favor certain perspectives over others, inadvertently reinforcing biases present in the data. As a result, users are fed information that perpetuates stereotypes, causing harm to underrepresented communities. To address these fairness issues, a number of ideas have also been proposed in the FL literature, as discussed further in Section 2.

In this paper, we establish an alignment between these two directions: we show that personalization techniques can also bring fairness benefits. Consider the concept of clustered FL, a personalized FL technique, as exemplified by the works of Ghosh et al. (2020); Nardi et al. (2022), where clients are organized into groups based on model similarities, so that knowledge is effectively aggregated within clusters of similar clients. We argue that this clustering can be seen as a means to foster a

model that is both more precise and more equitable. It effectively treats information originating from other clusters as noise, which, if left unaddressed, would have led to model divergence, potentially compromising both cluster-specific performance and fairness.

Furthermore, collaborative learning algorithms in essence have access to more "information" (data) from diverse clients. We argue that they can enhance both local accuracy and fairness, especially when dealing with imbalanced samples within each client, as in these scenarios issues of algorithmic unfairness can be attributed to under-representation of samples from the disadvantaged group in the data. Moreover, we identify cases where the sample distribution within clients is such that improving accuracy (over standalone learning) also promotes fairness; in these cases, the incorporation of personalization techniques into the FL paradigm may not only enhance local accuracy, but also yield improvements in local fairness, essentially providing a dual benefit.

To the best of our knowledge, this work is the first to identify the (unintended) fairness benefits of personalization techniques in federated learning. Most existing work primarily focus on refining algorithms to attain either improved local accuracy (the personalized FL literature), *or*, enhanced (global) fairness (the fair FL literature); we discuss related work from each direction in Section 2. In contrast, our work examines the influence of existing personalization techniques on group fairness through theoretical analysis and numerical experiments, and points to inherent features of collaborative learning and personalization that can advance fairness. Additionally, prior work points to the challenge of balancing the trade-off between fairness and accuracy of an algorithm, as enhancing fairness often comes at the cost of accuracy degradation (Gu et al., 2022; Ezzeldin et al., 2023). In contrast, we identify instances that federated learning in general, and personalized FL in particular, can improve *both* accuracy and fairness compared to standalone/independent learning.

**Summary of findings and contributions.** Our main findings and contributions are as follows:

1. We show that collaborative learning algorithms can enhance local fairness compared to standalone training, especially in the (commonly arising) situation where samples are imbalanced between two protected groups among clients (Section 4).
2. We numerically show that introducing personalization techniques can improve local fairness compared to a (non-personalized) FedAvg algorithm on real-world data (the Retiring Adult dataset Ding et al. (2021)) (Section 4) and synthetic data (Section 5).
3. We analytically show that, under certain conditions, personalization through clustering can lead to improved local fairness compared to a (non-personalized) FedAvg global model (Section 5).

## 2   RELATED WORKS

In this section, we present an overview of the existing personalization techniques and the methods for achieving fairness in the FL literature. We review additional related work in Appendix A.

**Personalized FL.** Existing literature can be categorized based on how personalization is achieved. *Model regularization:* Hanzely & Richtárik (2020); Sahu et al. (2018); Li et al. (2021) add a regularization term with mixing parameters to penalize the distance between the local and global models. In particular, Sahu et al. (2018) has a pre-set regularization parameter and allows for system heterogeneity, where each client can have a different amount of work to do. Li et al. (2021) consider improving accuracy, while being robust to data and model poisoning attacks, and fair. Similarly, T Dinh et al. (2020) formulate a bi-level optimization problem, which helps decouple personalized model optimization from learning the global model. Huang et al. (2021) propose the FedAMP algorithm which also introduce an additional regularization term, but differ from the previous works in that they encourage similar clients to collaborate more.
*Clustering:* Mansour et al. (2020) use a hypothesis-based clustering approach by minimizing the sum of loss over all clusters. Sattler et al. (2020) use the idea that cosine similarity between weight updates of different clients is highly indicative of the similarity of data distribution. Nardi et al. (2022) use a decentralized learning idea by exchanging local models with other clients to find the neighbor/group which has a high accuracy even using other clients' models. Zheng et al. (2022) learn a weighted and directed graph that indicates the relevance between clients. Ghosh et al. (2020) use a distributed learning idea by broadcasting all clients models to others, and collecting back the cluster identity from clients who can identify good performance when using others' models.
*Local fine-tuning:* Fallah et al. (2020) propose using a Model Agnostic Meta Learning (MAML) framework, where clients run additional local gradient steps to personalize the global model. Ari-

vazhagan et al. (2019); Jiang & Lin (2022) propose using deep learning models with a combination of feature extraction layers (base) and global/local head (personalization). Jiang & Lin (2022), inspired by Arivazhagan et al. (2019), further consider robustifying against distribution shifts.

In this paper, we investigate the fairness achieved by the clustering and local fine-tuning categories.

**Fairness in FL.** This literature, surveyed recently in Shi et al. (2023); Rafi et al. (2023), can also be categorized depending on the adopted notion of fairness as follows:

*Performance fairness:* This line of work measures fairness based on how well the learned model(s) can achieve uniform accuracy across all clients. Li et al. (2019a) propose the $q$-fair FL algorithm which minimizes the aggregate reweighted loss. The idea is that the clients with higher loss will be assigned a higher weight so as to encourage more uniform accuracy across clients. Li et al. (2021) further extend this by considering robustness and poisoning attacks; here, performance fairness and robustness are achieved through a personalized FL method. Zhang et al. (2021) aim to achieve small disparity in accuracy across the groups of client-wise, attribute-wise, and potential clients with agnostic distribution, simultaneously. Wang et al. (2021) discuss the (performance) unfairness caused by conflicting gradients. They detect this conflict through the notion of cosine similarity, and iteratively eliminate it before aggregation by modifying the direction and magnitude of the gradients.

*Social fairness:* This notion, which is also our notion of fairness in this paper, aims to minimize the disparity in decisions made across different demographic/protected groups. Abay et al. (2020) propose pre-processing (reweighting with differential privacy) and in-processing (adding fairness-aware regularizer) methods to mitigate biases while protecting the data privacy. Zhang et al. (2020) propose a fair FL framework consisting of a deep multi-agent reinforcement learning framework and a secure information aggregation protocol. They design the reward aligned with maximizing the global accuracy while minimizing the discrimination index between groups to overcome the accuracy and fairness trade-off challenge. Du et al. (2021) propose a fairness-aware agnostic FL framework to train a globally fair model with unknown testing data distribution or with domain adaptation through a kernel reweighting technique on both loss function and fairness constraints. Gálvez et al. (2021) mimic the centralized fair setting and introduce an algorithm to enforce group fairness in FL by extending the modified method of differential multiplier to empirical risk minimization with fairness constraints. Zeng et al. (2021); Ezzeldin et al. (2023) propose an in-processing approach to update the weight for each group based on the performance at each round, but they differ in the reweighting technique. Ezzeldin et al. (2023) propose fairness-aware aggregation by adjusting the weight based on how far the local fairness compared to global fairness, whereas Zeng et al. (2021) update the weights according to iteratively solving a bi-level optimization problem similar to the centralized fair learning algorithm. In contrast to all these works, we do not aim to impose a fairness constraint, but show that improved group social fairness (and a better fairness-accuracy tradeoff) can be achieved by personalization alone.

## 3 PROBLEM FORMULATION

In this study, we consider a FL scenario involving a total of $n$ clients, which can (potentially) be categorized into two clusters denoted as $c = \{\diamond, \square\}$ based on similarities in their data distributions. The number of clients in each cluster is denoted as $|\diamond|$ and $|\square|$, where a fraction $p$ of the clients belong to the cluster $\diamond$, such that $|\diamond| = np$ and $|\square| = n(1 - p)$. Each client $i$ is tasked with a binary classification problem, where data points are randomly drawn from the joint distribution $f_g^{y,c}$. A data point $z = (x, y, g)$ comprises a feature or score denoted as $x \in \mathbb{R}$ and a true label denoted as $y \in \{0, 1\}$. In this context, $y = 1$ and $y = 0$ represent qualified and unqualified agents, respectively. Additionally, each data point is associated with a group membership denoted as $g \in \{a, b\}$, determined by demographic or protected attributes (e.g., race or gender). To classify the data points, each client employs threshold-based, binary classifiers $h_\theta(x) : \mathcal{X} \to \{0, 1\}$.[1] Here, $\theta$ represents the decision threshold for each client. Any data point belonging to group $g$ with a feature value $x \geq \theta_g$ is assigned the label 1. The objective for each client may involve minimizing the classification error, maximizing profit, or pursuing other relevant goals. For instance, the formula

---

[1]Existing work in Corbett-Davies et al. (2017, Thm 3.2) and Raab & Liu (2021) show that threshold classifiers can be optimal if multi-dimensional features can be properly mapped into a one-dimensional space; we could consider using the last layer outputs from a neural network as this one-dimensional representation.

for minimizing classification error can be expressed as follows, where $\alpha_g^y$ represents the fraction of data in group $g$ with label $y$ and $gr_g$ represents the fraction of data in group $g$:

$$\min_{\theta} \sum_{g \in \{a,b\}} gr_g \left( \alpha_g^1 \int_{-\infty}^{\theta} f_g^1(x)\mathrm{d}x + \alpha_g^0 \int_{\theta}^{\infty} f_g^0(x)\mathrm{d}x \right) \tag{1}$$

**FL algorithms.** The FedAvg algorithm (McMahan et al., 2017) operates by establishing connections between a central server and $n$ distinct clients. The objective is to learn a global model $w$ that minimizes the average loss across all clients. During each communication round $t$, individual client $i$ receives the global model from the previous round $w^{t-1}$, performs a local update, and sends back the updated model $w_i^t$ to the server. The server then (weighted) aggregates all the received local models to create a new global model for the next round. In contrast, the clustered FL algorithm (Ghosh et al., 2020) introduces additional steps beyond the FedAvg approach. It begins by clustering clients based on model similarities. Once the clients are organized into clusters, the server (weighted) aggregates the local models from each cluster, forming a cluster-specific model to be used for the next round of training in that cluster. Similarly, the MAML FL algorithm (Fallah et al., 2020) also takes an extra step beyond the FedAvg approach. It updates the received global model by running additional gradient steps locally, and then the server (weighted) aggregates the updated models from each client used for the next round of training.

**Fairness metric:** Considering the sensitive demographic information $g$ in each data point within our problem setting, our goal is to ensure fairness across different demographic groups. To measure fairness, we adopt the statistical parity fairness metric (Dwork et al., 2012; Ezzeldin et al., 2023), which emphasizes equalizing the positive classification rate for each group. Formally:
$$ASPD = |Pr(\hat{y} = 1|g = a) - Pr(\hat{y} = 1|g = b)|$$
For the sake of simplicity in our theoretical analysis, we make the assumption that clients within the same cluster are identical. Consequently, the $ASPD$ calculated for each client can also serve as a representation of the local fairness performance for the cluster to which the client belongs. Given that the local fairness performance for an algorithm can be computed as a weighted sum of the local fairness performance from each cluster, we also introduce the concept of *average* local fairness.

**Definition 1.** *Let $\Delta(\theta)$ denote the cluster-wise average local statistical parity fairness gap under different models' optimal solution $\theta$, such that*
$$\Delta(\theta) = pASPD_{\diamond} + (1-p)ASPD_{\square}$$

## 4 NUMERICAL EXPERIMENTS

In this section, we compare the average local statistical parity fairness achieved by two personalized FL algorithms (clustered FL Ghosh et al. (2020) from the clustering category, and MAML Fallah et al. (2020) from the local fine-tuning category), against FedAvg and standalone learning (where each client learns by itself, and there is no collaborative training).

We show experiments on the pre-processed Retiring adult dataset (Ding et al., 2021). The dataset consists of census data collected from all 50 states in the US and Puerto Rico. In our context, each individual state is a client within the FL framework. Each data sample includes multi-dimensional features $x$ (e.g., age, education, citizenship, etc.), a true label (denoted as $y$), and a protected attribute (referred to as $g$, e.g., gender, race). To provide a more clear comparison of the effects of personalization, we have manually scaled the feature set ($x$) by 60% for the states with IDs $\{1, 10, 20, 30, 40, 50\}$; this exacerbates the data heterogeneity. We focus on two binary classification tasks: Employment (ACSEmployment) and Income (ACSIncome), and employ a two-layer neural network for both tasks. Each client is provided with 1000 training samples and 2000 testing samples. The local accuracy of FedAvg is computed as the average across all 51 clients when the common global model is applied to their local datasets. For the clustered FL algorithm, this performance metric is derived based on the corresponding cluster model for each client, and for MAML FL, it is from the local model obtained after local fine-tuning. The values reported are the averages from 5 runs.

Further details regarding the datasets and models, and large (full) figures can be found in Appendix B. We also provide additional experiments on synthetic datasets in Section 5 and Appendix D.

## 4.1 DATASETS WITH IMBALANCED GROUPS

We first consider the ACSEmployment dataset with race as the protected attribute. Fig 1(a) shows the fraction of samples in each group/label, from several states, highlighting an imbalance between samples from the White and Non-White groups. This is further evident in Figure 1(b), which shows that most states have only ∼ 10% qualified (label 1) samples from the Non-White group, in contrast to ∼ 35% qualified samples from the White group.

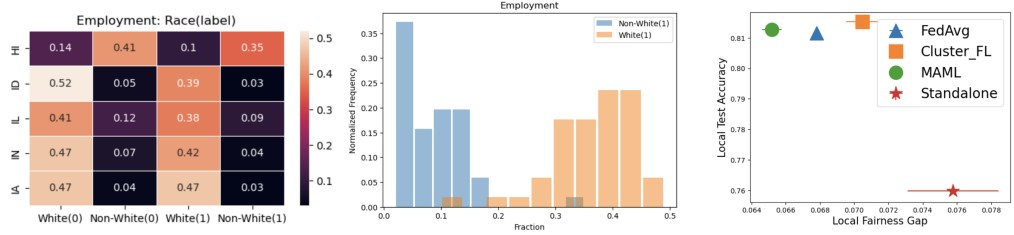

(a) Fraction of samples  (b) Normalized sample frequency  (c) Local accuracy vs. fairness gap

Figure 1: Experiments using ACSEmployment dataset with imbalanced groups (Race)

From Fig 1(c), we can see that all collaborative training algorithms (FedAvg, MAML, Clustered FL) have better local fairness (smaller gap) compared to standalone learning. This is expected because each client has limited samples in their disadvantaged group, leading to poorly trained models with high local fairness gaps (and low accuracy). However, collaborative training in essence has access to more data, improving both metrics. For the same reason, the clustered FL algorithm which partitions clients into two clusters, has (slightly) worse local fairness performance compared to FedAvg. In contrast, the MAML (fine-tuning) algorithm, which effectively sees the global dataset when training the global model, has both better local fairness compared to the FedAvg algorithm, indicating that the introduction of personalization techniques can improve both local accuracy and fairness.

## 4.2 DATASETS WITH BETTER-BALANCED GROUPS

We next conduct experiments with more balanced groups. We again consider the ACSEmployment dataset, but with gender as the protected attribute. We can see from Fig 2(a) that the fractions of samples are more evenly distributed across groups and labels. Furthermore, Figure 2(b) confirms that many states exhibit similar sample fractions between male and female groups, as indicated by the greater overlap of the bars compared to Figure 1(b).

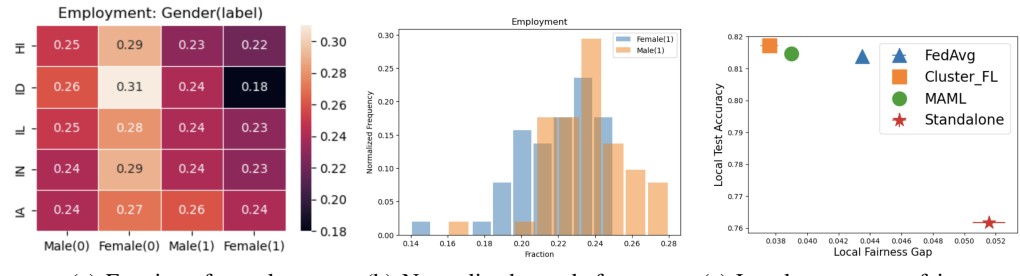

(a) Fraction of samples  (b) Normalized sample frequency  (c) Local accuracy vs. fairness gap

Figure 2: Experiments using ACSEmployment dataset with better-balanced groups (Gender)

We first notice from Fig 2(c) that all collaborative training algorithms still have better local fairness performance compared to standalone learning. Furthermore, we observe that both Clustered FL and MAML FL achieve both better local accuracy and local fairness compared to FedAvg. This is because for each client, due to similarity of the data for the male and female groups (as seen in Figure 2(b)) the objective of maximizing local accuracy is aligned with reducing the fairness gap. Therefore, bringing the personalization techniques into the FL paradigm can also improve local fairness performance for free.

We also conduct experiments on another task, the ACSIncome dataset, again with gender as the protected attribute. We observe from Fig 3(a) that the fraction of samples is comparable across groups for unqualified (label 0) data, but differs for qualified (label 1). From Fig 3(c), we observe that this

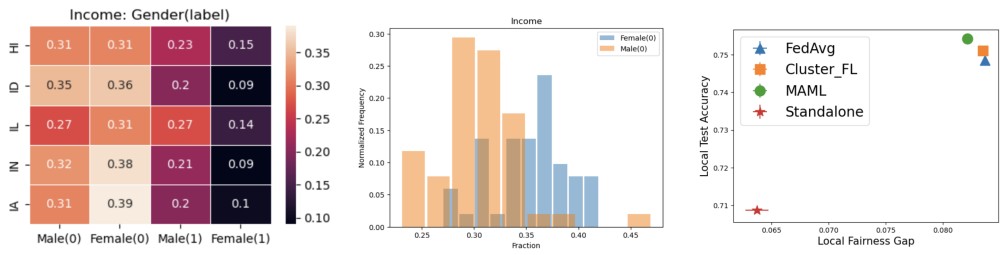

(a) Fraction of samples (b) Normalized sample frequency (c) Local accuracy vs. fairness gap

Figure 3: Experiments using ACSIncome dataset with gender as protected attribute

time, all collaborative training algorithms have *worse* local fairness compared to standalone learning. Furthermore, we observe that the personalized FL algorithms slightly improve local fairness compared to FedAvg, but this is less significant compared to the results in Fig 2(c). We believe this is due to the objective of maximizing local accuracy not aligning with fairness. In other words, collaborative training algorithms maximize the local accuracy performance while sacrificing the local fairness performance. However, as we discussed in Fig 1(c), collaborative training could have a chance to improve local fairness performance by seeing more information from diverse clients. Therefore, a combined effect makes the overall local fairness advantages less significant.

### 4.3 COMPARISON WITH FAIR FL ALGORITHMS

A natural question that may arise is why we do not utilize an existing fair FL algorithm to improve fairness, as these might offer superior fairness compared to a personalized FL algorithm. Indeed, if one only focuses on improving (global) fairness, choosing a dedicated Fair FL algorithm could be the best choice. However, here we point to the additional (cost-free) local fairness improvements achieved through the incorporation of personalization in FL. Our finding also suggest that this leads to a more favorable trade-off between fairness and accuracy.

To show this, we consider the existing FairFed algorithm (Ezzeldin et al., 2023), which adjusts the aggregation weight according to a fairness metric with the goal of improving (global) fairness. We also consider an extension of the algorithm, FairFed-AF, which adjusts the aggregations weights according to both accuracy and fairness metrics. The experiments in Fig 4 use the ACSEmployment dataset with gender as the protected attribute. We observe that fair FL algorithms can achieve the best fairness among all other algorithms. However, they have worse local accuracy compared to other collaborative training algorithms, as they focus (at least partially) on improving fairness and not accuracy.

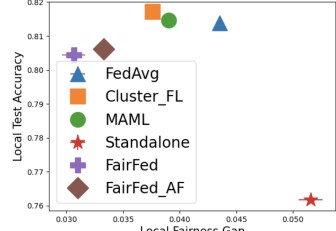

Figure 4: Comparison between FedAvg, fair FL, personalized, and standalone training.

## 5 THEORETICAL ANALYSIS AND ILLUSTRATIVE EXAMPLES

To support and validate our findings from the numerical experiments, in this section, we provide analytical results showing that clustered FL can lead to better fairness (measured by $ASPD_c$) than FedAvg under certain conditions. We provide additional intuition with numerical experiments showing the impact of cluster sizes on the (cluster-wise) average local fairness (measured by $\Delta(\theta)$). We provide additional examples on synthetic data in Appendix D.

**Fairness comparison setup:** Let $\theta^{*,c}$, $\theta^{FA}$, and $F_g^{y,c}$ represent the optimal solutions for cluster $c$ in the clustered FL, the FedAvg algorithm, and the cumulative distribution functions (CDFs) of the data distributions of cluster $c = \{\Diamond, \Box\}$, label $y \in \{0, 1\}$, and group $g \in \{a, b\}$, respectively. Additionally, $\alpha_g^{y,c}$ denotes the fraction of group $g$ data in cluster $c$ with label $y$. To compare the local fairness performance between the clustered FL and FedAvg algorithms, we for simplicity assume that clients within the same cluster are identical. According to this assumption and the FedAvg aggregation technique, the optimal solution for the FedAvg algorithm would lie between $\theta^{*,\Diamond}$ and

$\theta^{*,\square}$. Without loss of generality, we assume that $\theta^{*,\diamond} < \theta^{*,\square}$. At the optimal solutions, for cluster $\diamond$, the local fairness can be expressed as follows:

$$Pr(\hat{y} = 1 | g = a, c = \diamond) = Pr(\hat{y} = 1, y = 1 | g = a, c = \diamond) + Pr(\hat{y} = 1, y = 0 | g = a, c = \diamond)$$

$$= \alpha_a^{1,\diamond} \int_{\theta^{*,\diamond}}^{\infty} f_a^{1,\diamond}(x)\mathrm{d}x + \alpha_a^{0,\diamond} \int_{\theta^{*,\diamond}}^{\infty} f_a^{0,\diamond}(x)\mathrm{d}x = 1 - \alpha_a^{1,\diamond} F_a^{1,\diamond}(\theta^{*,\diamond}) - \alpha_a^{0,\diamond} F_a^{0,\diamond}(\theta^{*,\diamond})$$

Therefore, the $ASPD_\diamond = |\alpha_b^{1,\diamond} F_b^{1,\diamond}(\theta^{*,\diamond}) + \alpha_b^{0,\diamond} F_b^{0,\diamond}(\theta^{*,\diamond}) - \alpha_a^{1,\diamond} F_a^{1,\diamond}(\theta^{*,\diamond}) - \alpha_a^{0,\diamond} F_a^{0,\diamond}(\theta^{*,\diamond})|$, and the $\Delta(\theta^{*,c})$ for the clustered FL algorithm is:

$$\Delta(\theta^{*,c}) = p\Big(|\alpha_b^{1,\diamond} F_b^{1,\diamond}(\theta^{*,\diamond}) + \alpha_b^{0,\diamond} F_b^{0,\diamond}(\theta^{*,\diamond}) - \alpha_a^{1,\diamond} F_a^{1,\diamond}(\theta^{*,\diamond}) - \alpha_a^{0,\diamond} F_a^{0,\diamond}(\theta^{*,\diamond})|\Big)$$

$$+ (1-p)\Big(|\alpha_b^{1,\square} F_b^{1,\square}(\theta^{*,\square}) + \alpha_b^{0,\square} F_b^{0,\square}(\theta^{*,\square}) - \alpha_a^{1,\square} F_a^{1,\square}(\theta^{*,\square}) - \alpha_a^{0,\square} F_a^{0,\square}(\theta^{*,\square})|\Big)$$

Similarly, the cluster-wise average local fairness for the FedAvg algorithm could be written in the same manner. Considering the expressions presented above, we can observe that the first term in $\Delta(\theta^{*,c})$, weighted by the cluster size parameter $p$, represents the statistical parity fairness gap within cluster $\diamond$ evaluated at its optimal decision threshold. Similarly, the second term, weighted by $1-p$, reflects the same phenomenon but within cluster $\square$. Consequently, the following propositions elucidate the impact of transitioning from the clustered FL solution, denoted as $\theta^{*,c}$, to the FedAvg solution, represented as $\theta^{FA}$, on individual clusters. In essence, if the FedAvg solution $\theta^{FA}$ causes a deterioration in fairness for both clusters in comparison to the clustered FL solutions, the cluster-wise average statistical parity fairness gap $\Delta(\theta^{FA})$ will consequently exhibit a worse local fairness performance than the clustered FL solutions. However, if the FedAvg solution $\theta^{FA}$ results in worsened fairness for one cluster while simultaneously enhancing fairness for the other cluster, the cluster-wise average fairness performance under the clustered FL solution could still be improved, provided that the cluster experiencing an improvement in fairness has a larger cluster size.

For simplicity, the following propositions specifically focus on the scenario within cluster $\diamond$, where samples are drawn independently and identically from corresponding Gaussian distributions. The analysis could also be applied to the cluster $\square$ in the same manner. These Gaussian distributions exhibit equal variances denoted as $\sigma$ but possess different mean estimates $\mu_g^y$. In this context, we assume that $\mu_b^0 \le \mu_a^0 \le \mu_b^1 \le \mu_a^1$. Furthermore, both propositions investigate cases where there is an equalized gap between mean estimates such that $\mu_a^1 - \mu_a^0 = \mu_b^1 - \mu_b^0$. To provide a comprehensive understanding, we also offer illustrative examples under scenarios where the gap is not equalized in Appendix D. Proposition 1 considers an assumption of equalized label participation rates, where $\alpha_g^y$ are the same across labels and groups. In contrast, Proposition 2 relaxes this assumption. In the following analysis, we omit the cluster notation when it is clear from the context. Detailed proofs are presented in Appendix C.

**Proposition 1.** *Consider the problem setup and fairness comparison setup in Section 3 and 5, within one cluster (e.g., $\diamond$), we assume samples from label $y$ and group $g$ follow the Gaussian distributions with equal variance $\sigma$ and the corresponding mean estimates $\mu_g^y$. Furthermore, we consider that each label distribution of the disadvantaged group ($g = b$) is located on the left-hand side of the corresponding label distribution of the advantaged group ($g = a$) with equal distance such that $\mu_b^0 \le \mu_a^0 \le \mu_b^1 \le \mu_a^1$ and $\mu_a^1 - \mu_a^0 = \mu_b^1 - \mu_b^0$. For simplicity, we assume the size of samples with different labels $y$ is balanced in two groups such that $\alpha_g^y$ are the same across labels and groups. Let $\theta^{*,\diamond}$ be the optimal decision threshold for cluster $\diamond$ obtained by solving 1. W.o.l.g., we consider $\theta^{*,\diamond} < \theta^{*,\square}$, the optimal decision threshold for the other cluster $\square$. Then, if the condition $\exp(-\frac{(\mu_b^1 - \mu_a^0)^2}{8\sigma^2})(\mu_b^1 - \mu_a^0) \ge \exp(-\frac{(\mu_a^1 - \mu_b^0)^2}{8\sigma^2})(\mu_a^1 - \mu_b^0)$ holds, then there exist a cluster size weight $p$ such that the FedAvg solution $\theta^{FA} := p\theta^{*,\diamond} + (1-p)\theta^{*,\square}$ will make the cluster $\diamond$ unfairer.*

**Proposition 2** (Extension of Prop. 1)**.** *Consider the setup in Proposition 1, we relax the balanced label participation rate in two groups such that $\alpha_g^y$ could be different. Let $\theta^{*,\diamond}$ and $\bar{\theta}$ be the optimal decision threshold for cluster $\diamond$ obtained by solving 1 and the algorithmic average of $\mu_g^y$. Then, when the majority of samples are labeled 1 in two groups (e.g., $\alpha_g^1 \ge \alpha_g^0$), or when $gr_b \ge gr_a$ and the majority of samples are labeled 1 in one group where the other group has a better balance in label but the majority of samples are labeled differently (e.g., $\alpha_a^1 > \alpha_b^0 > \alpha_b^1 > \alpha_a^0$), if the condition $\alpha_a^0 \exp(\frac{(\bar{\theta} - \mu_a^0)^2}{-2\sigma^2})(\bar{\theta} - \mu_a^0) - \alpha_b^1 \exp(\frac{(\bar{\theta} - \mu_b^1)^2}{-2\sigma^2})(\bar{\theta} - \mu_b^1) \ge \alpha_b^0 \exp(\frac{(\bar{\theta} - \mu_b^0)^2}{-2\sigma^2})(\bar{\theta} - \mu_b^0) - \alpha_a^1 \exp(\frac{(\bar{\theta} - \mu_a^1)^2}{-2\sigma^2})(\bar{\theta} - \mu_a^1)$ holds, then there exist a cluster size weight $p$ such that the FedAvg solution $\theta^{FA} := p\theta^{*,\diamond} + (1-p)\theta^{*,\square}$ will make the cluster $\diamond$ unfairer.*

**Numerical illustration.** We now conduct numerical experiments to illustrate the findings in Proposition 1 and 2. The results are presented in Table 1 and 2. We proceed as follows: 10000 random samples are drawn from Gaussian distribution for each group $g \in \{a, b\}$ with mean $\mu_g^y$ and standard deviation $\sigma$. The number of qualified ($y = 1$) and unqualified ($y = 0$) samples in each group is proportional to the label participation rate $\alpha_g^y$. Although the samples were generated in a consistent manner across different parameter settings, we assumed an optimal decision threshold $\theta^{*,\square} = 8$ for cluster $\square$, as both Propositions investigate the fairness performance for cluster $\diamond$. In Table 1, we consider the scenario where $\alpha_g^y = 0.5 \ \forall g, y$. In contrast, different values of $\alpha_g^y$ are applied in Table 2. Both results in Table 1 and 2 consider an equalized group rate such that $gr_a = gr_b$ and an equalized gap between mean estimates. We conducted each experiment 5 times and reported the average value of the fairness gap. Additional comparative experiments, including cases with unequalized group rates and gaps between mean estimates, are presented in Appendix D.

From Table 1, we can find that it offers crucial insights into the conditions required for Proposition 1 to hold. For fixed mean estimates $\mu_g^y$ (rows 1-2), we observe that smaller values of $\sigma$ are preferable to satisfy the specified conditions. Similarly, for fixed $\sigma$ (row 1, 3 and row 2, 4), larger differences between $\mu_g^1$ and $\mu_g^0$ are advantageous in fulfilling the conditions outlined in Proposition 1. This observation becomes intuitive at the extreme cases where the $\sigma$ is sufficiently small or the distances between $\mu_g^1$ and $\mu_g^0$ are sufficiently large. In these extreme scenarios, it becomes possible to consider the samples as linearly separable. Therefore, the optimal decision threshold $\theta^{*,\diamond}$ could achieve a perfect classification as well as perfect fairness. As a result, the FedAvg solution $\theta^{FA}$ deviated from the optimal solution will lead to a worse performance in both accuracy and fairness.

Table 1: Cluster $\diamond$ fairness performance with equalized gap and label rate

| Distribution $(\mu_a^1, \mu_a^0, \mu_b^1, \mu_b^0, \sigma)$ | Condition satisfied | $ASPD_\diamond(\theta^{*,\diamond})$ | $ASPD_\diamond(\theta^{FA})$ $p = \frac{2}{3}$ | $p = \frac{1}{2}$ |
|---|---|---|---|---|
| $(7, 4, 6, 3, 1)$ | Yes | 0.1359 | 0.1814 ↑ | 0.1945 ↑ |
| $(7, 4, 6, 3, 2)$ | No | 0.1499 | 0.1417 ↓ | 0.1315 ↓ |
| $(7, 5, 6, 4, 1)$ | No | 0.1747 | 0.1638 ↓ | 0.1511 ↓ |
| $(8, 3, 6, 1, 2)$ | Yes | 0.1866 | 0.1968 ↑ | 0.2033 ↑ |

Table 2 reveals insights regarding the influence of label distribution on fairness performance and decision thresholds. Specifically, when the majority of samples in both groups are labeled as 1 (rows 1-2), the optimal decision threshold ($\theta^{*,\diamond}$) shifts leftward compared to the balanced scenario. However, since $\theta^{*,\diamond} < \theta^{*,\square}$, the FedAvg solution $\theta^{FA}$ will be greater than $\theta^{*,\diamond}$. Moreover, based on the expression of the fairness gap $\Phi(\theta) = \alpha_b^0 F_b^0(\theta) + \alpha_b^1 F_b^1(\theta) - \alpha_a^0 F_a^0(\theta) - \alpha_a^1 F_a^1(\theta)$ and our assumptions, we can find that $\theta$ will have even larger fairness gap when it is shifted to the right because the rate of change in $F_b^1(\theta)$ is larger than that of $F_a^1(\theta)$, while the rate of change in $F_g^0(\theta)$ is dominated by the terms $F_g^1(\theta)$, who have heavier weights, especially when the standard deviation $\sigma$ is small. Another intriguing observation is that in cases where the majority of samples have different labels (row 3), the FedAvg solution ($\theta^{FA}$) yields worse fairness performance when $p = 2/3$ but not when $p = 1/2$. This outcome aligns with expectation, as the fairness gap approaches 0 as $\theta \to \infty$ in the definition of $\Phi(\theta)$. As the cluster size weight $p$ enlarges (resp. reduces), the FedAvg solutions $\theta^{FA}$ will be closer to the $\theta^{*,\diamond}$ (resp. $\theta^{*,\square}$), which also indicates a significant role played by the cluster size weight ($p$) in shaping the overall cluster-wise average fairness performance, especially when assessing the impact of FedAvg solutions on two clusters, $\diamond$ and $\square$.

Table 2: Cluster $\diamond$ fairness performance with equalized gap

| Distribution $(\mu_a^1, \mu_a^0, \mu_b^1, \mu_b^0, \sigma)$ | Label rate $(\alpha_a^1, \alpha_a^0, \alpha_b^1, \alpha_b^0)$ | Condition satisfied | $ASPD_\diamond(\theta^{*,\diamond})$ | $ASPD_\diamond(\theta^{FA})$ $p = \frac{2}{3}$ | $p = \frac{1}{2}$ |
|---|---|---|---|---|---|
| | $(0.7, 0.3, 0.6, 0.4)$ | Yes | 0.2062 | 0.2832 ↑ | 0.3041 ↑ |
| $(7, 4, 6, 3, 1)$ | $(0.6, 0.4, 0.7, 0.3)$ | Yes | 0.0453 | 0.1433 ↑ | 0.1961 ↑ |
| | $(0.7, 0.3, 0.4, 0.6)$ | Yes | 0.3797 | 0.3962 ↑ | 0.3676 ↓ |
| | $(0.6, 0.4, 0.3, 0.7)$ | No | 0.3797 | 0.3598 ↓ | 0.3189 ↓ |

In Table 3, we delve into the cluster-wise average statistical parity fairness gap $\Delta(\theta)$ achieved with different decision thresholds (optimal clustered FL solutions $\theta^{*,c}$ and FedAvg solutions $\theta^{FA}$). In this investigation, we maintain the parameters in cluster $\square$ while varying those in cluster $\diamond$ to assess fairness impact. We assume equalized group rates for this experiment, with additional experiments

considering unequalized group rates provided in Appendix D. From the results in Table 3, we can find that when both conditions are not satisfied (rows 5-6), there is a cluster size weight $p$ such that the FedAvg solutions would lead to better fairness performance for each cluster, consequently yielding a lower cluster-wise average fairness gap. However, when only one cluster satisfies the condition, meaning that there is a $p$ such that the FedAvg solutions would only make one cluster unfairer (rows 1-2), we could see that a relatively small $p$ would let the clustered FL yield a better fairness performance. Nevertheless, when $p$ is sufficiently small, the FedAvg solutions will again have superior fairness performance than the clustered FL solutions, similar to the results in rows 3-4. Essentially, for each cluster $c$, there exists a range $(p_{low}^c, p_{high}^c)$ such that, within this range, FedAvg solutions result in worse fairness performance compared to clustered FL solutions. Consequently, for any $p \in \cap_c(p_{low}^c, p_{high}^c)$, clustered FL solutions yield a superior cluster-wise average statistical parity fairness performance relative to FedAvg solutions.

Table 3: Cluster-wise average fairness performance with equalized gap

| Distribution $\diamond : (\mu_a^1, \mu_a^0, \mu_b^1, \mu_b^0, \sigma)$ $\square : (\mu_a^1, \mu_a^0, \mu_b^1, \mu_b^0, \sigma)$ | Label rate $(\alpha_a^1, \alpha_a^0, \alpha_b^1, \alpha_b^0)$ $(\alpha_a^1, \alpha_a^0, \alpha_b^1, \alpha_b^0)$ | Condition satisfied | $p$ | $\Delta(\theta^{*,c})$ | $\Delta(\theta^{FA})$ |
|---|---|---|---|---|---|
| (7, 4, 6, 3, 2) | (0.5, 0.5, 0.5, 0.5) | No | 4/5 | 0.147 | 0.144 ↓ |
| (10, 7, 9, 6, 1) | (0.5, 0.5, 0.5, 0.5) | Yes | 1/3 | 0.141 | 0.160 ↑ |
| (7, 4, 6, 3, 2) | (0.8, 0.2, 0.3, 0.7) | Yes | 1/4 | 0.205 | 0.199 ↓ |
| (10, 7, 9, 6, 1) | (0.5, 0.5, 0.5, 0.5) | Yes | 1/2 | 0.274 | 0.277 ↑ |
| (7, 4, 6, 3, 2) | (0.5, 0.5, 0.5, 0.5) | No | 1/3 | 0.254 | 0.222 ↓ |
| (10, 7, 9, 6, 1) | (0.7, 0.3, 0.4, 0.6) | No | 2/3 | 0.202 | 0.166 ↓ |

## 6 CONCLUSION, LIMITATIONS, AND EXTENSIONS

We studied the (unintended) fairness benefits of personalization techniques in federated learning. Instead of relying on a dedicated fair FL algorithm, we can improve local fairness and have a potentially more desirable fairness-accuracy tradeoff, through the introduction of personalization techniques alone. We find that when samples are imbalanced, collaborative training (whether personalized or not) can yield both better accuracy and better fairness than individual training. Moreover, when the objectives of accuracy and fairness are aligned (as we identify in real-world datasets), introducing personalization can also improve local fairness for free. Our analytical results provide support for these findings under some conditions.

**Distribution assumption.** It's essential to note that our analysis relies on the assumption that samples are drawn IID from the Gaussian distribution. Therefore, the derived conditions pertain specifically to this distribution. An intriguing avenue for future work involves conducting a distribution-agnostic investigation of fairness impacts. Furthermore, we've provided numerical support for scenarios with unequalized gaps between mean estimates in Appendix D. Extending the corresponding theoretical analysis would be a valuable pursuit. The exploration of the existence of $p_{low}$ and $p_{high}$, as well as deriving closed-form expressions for these values, represents another intriguing direction. **Fairness.** In this paper, our focus centers on local statistical parity fairness performance. However, other fairness notions such as equality of opportunity and equalized odds, as well as the constrained optimization problem remain unexplored and warrant further investigation. A comprehensive analysis, accompanied by numerical results, would also be worth exploring. **Personalization techniques.** While this study treats clustering as a form of personalization technique, our analytical exploration primarily concerns the fairness impact between clustered FL algorithms and FedAvg algorithms. In our experiments, we also explore the fairness impact using the MAML FL algorithm. Extending the theoretical analysis to encompass other types of personalized FL algorithms constitutes a promising avenue. Furthermore, investigating the impact of fairness performance using a broader spectrum of personalized FL algorithms is a future direction. **FL setting.** Our study focuses on a setting with two clusters, where clients within the same cluster share identical data distribution and label fractions. Extending this framework to include multiple clusters and acknowledging client heterogeneity within each cluster is a straightforward extension. Additionally, allowing for drop-out clients, a scenario not considered in the current FL setting, presents another avenue for exploration.

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
