# OpenReview forum: "Enhancing Group Fairness in Federated Learning through Personalization"
_ICLR.cc/2024/Conference — Submitted to ICLR 2024_

### Official Review · Reviewer_gUoU · 2023-10-30

**Soundness:** 2 fair
**Presentation:** 1 poor
**Contribution:** 2 fair
**Rating:** 3
**Confidence:** 4

**Summary:**

This paper investigates the impact of personalization techniques in the FL paradigm on local (group) fairness of the learned models, and show that personalization techniques can also lead to improved fairness. In addition,
We establish this effect through numerical experiments comparing two types of personalized FL algorithms against the baselines and elaborate on the reasons behind improved fairness using personalized FL methods. What’s more, they further provide analytical support under certain conditions.

**Strengths:**

S1. The paper through the introduction of personalization techniques alone can improve local fairness and has a potentially more desirable fairness-accuracy tradeoff, which is important.
S2. They have provided theory analytics.

**Weaknesses:**

W1. The motivation for Formula 1 is expected with clear explanation.
W2. It is rather abrupt to say “Consequently, clients with heterogeneous datasets may encounter local accuracy degradation”. Please elaborate on it, preferably with an example.
W3. There is no relevant pseudocode in this article.
W4. The icon of figure1(c) is inconsistent.

**Questions:**

Q1. This sentence “It effectively treats information originating from other clusters as noise, which, if left unaddressed, would have led to model divergence, potentially compromising both cluster-specific performance and fairness.” What does this mean?
Q2. Is the algorithm in this article only for binary classification? Can it be adapted to more complex tasks?
Q3. Please provide a detailed explanation of all formulas in the "Fairness metric" chapter.

---

> ### Author Response · Authors · 2023-11-21
> **Thank you for your valuable suggestions and comments. We address the questions in the comments below.**
>
> Thank you for your valuable comments and suggestions.
>
> **Major comment 1**: Thank you for your comment on the motivation for Formula 1. In FL, each local client possesses its unique objective function $F_i$, and the FedAvg algorithm finds the optimal solution over $n$ clients such that $\arg\min_{\theta} F(\theta) := 1/n\sum_{i}F_i(\theta)$, where $F_i(\theta) := E_{(x,y)}\{L_i(\theta,x,y)\}$. Formula 1 represents the misclassification error where the first and second terms are the false negative and false positive rates under the decision threshold $\theta$.
>
> **Major comment 2**: Thank you for your careful reading. Regarding the degradation in local accuracy performance with heterogeneous data, one of the main reasons is the discrepancy between the local optima and the global optima for each client. As also illustrated in [1,2,3], when learning on heterogeneous data, the accuracy of FedAvg is significantly reduced because of client drift [4]. Given the data heterogeneity and the limited local data available, clients struggle to accurately estimate the entire data population, especially in cases of highly heterogeneous data. Consequently, this leads to a degradation in local accuracy.  To elaborate further, we could consider a simple example with two significantly different clusters in the clustered FL framework.  If we were to apply the FedAvg algorithm directly, then for each cluster, the information/knowledge obtained from the other cluster would hurt its performance. Conversely, employing the clustered FL algorithm effectively separates these clusters, allowing each to optimize for cluster-specific performance and fairness independently (treating information/knowledge coming outside from its cluster as noise).
>
> [1] Li, Tian, et al. "Federated optimization in heterogeneous networks." Proceedings of Machine learning and systems 2 (2020): 429-450.
>
> [2] Tan, Alysa Ziying, et al. "Towards personalized federated learning." IEEE Transactions on Neural Networks and Learning Systems (2022).
>
> [3] Yang, He, et al. "FedRich: Towards efficient federated learning for heterogeneous clients using heuristic scheduling." Information Sciences 645 (2023): 119360.
>
> [4] Karimireddy, Sai Praneeth, et al. "Scaffold: Stochastic controlled averaging for federated learning." International conference on machine learning. PMLR, 2020.
>
> **Major comment 3**: Thank you for your comment on the binary classification setting. You are correct that in this article we only investigate the fairness performance between two protected groups (e.g., Male/Female; White/Non-white) for clarity and comparability. As you have noticed, we could also adapt it to accommodate multi-class predictions. For example, there are nine different races for both ACSEmployment and ACSIncome tasks. We could investigate fairness impacts among multiple groups. However, the primary reason for not considering this aspect in our paper is the significant challenge posed by the notably small sample sizes for individual underprivileged groups (e.g., Black or African American, American Indian, Asian, etc.), unless considered collectively (e.g., Non-white).
>
> **Major comment 4**: Thank you for your careful reading. The ASPD refers to the Absolute Statistical Parity Difference. The statistical parity fairness is also known as the demographic parity. ASPD quantifies disparities in positive predictions ($\hat{y}=1$) between two groups ($a,b$). A smaller ASPD signifies a fairer model, indicating that positive prediction rates between the two groups closely align. In our assumption that clients within the same cluster are identical, the ASPD for the client could also serve to represent the cluster to which the client belongs. In scenarios with multiple clusters, as in our analysis with two clusters, we employ $\Delta(\theta)$ to measure the cluster-wise average ASPD, weighted by the cluster size $p$, where $\theta$ denotes the decision threshold for the corresponding algorithm.
>
> **Minor comment 1**: Thank you for your careful reading. Since we only investigate the fairness impact achieved through the introduction of the personalization technique in the FL framework (e.g., clustered FL, MAML FL), we do not present pseudocodes for existing well-known algorithms.

---

### Official Review · Reviewer_9xvK · 2023-10-31

**Soundness:** 2 fair
**Presentation:** 2 fair
**Contribution:** 2 fair
**Rating:** 3
**Confidence:** 4

**Summary:**

The paper proposes to explore the intersection of personalization techniques in federated learning (FL) with the goal of improving fairness.

**Strengths:**

It argues that the collaboration inherent in FL can enhance both local accuracy and fairness, especially when dealing with imbalanced samples and offers an empirical and theoretical foundation to support its claims. The authors also provide detailed theoretical proposition and analytical results to support the idea.

**Weaknesses:**

However,  while the work touches on an interesting aspect of federated learning, there are critical shortcomings and limitations.

1.	Lack of Novelty and Significance: The paper attempts to align personalization techniques with fairness benefits in FL. While the authors have shown that is it a valid and relevant research direction, the authors do not present a novel or significant contribution to the field. The concepts of clustered FL and personalized FL used in this paper is proposed by prior research works such as Ghosh et al. (2020) and Nardi et al. (2022). As a result, this work appears to be a reiteration or extension of established ideas, but it does not come across as a breakthrough or innovative approach.

2.	Limitation of Assumptions: The theoretical propositions make several assumptions, such as equalized label participation rates and balanced label distribution. While these assumptions are necessary for the theoretical analysis, they might not fully represent real-world scenarios, limiting the generalizability of the findings.

**Questions:**

See above

---

> ### Author Response · Authors · 2023-11-21
> **Thank you for your valuable suggestions and comments. We address the questions in the comments below.**
>
> Thank you for your valuable comments and suggestions.
>
> **Major comment 1**: Thank you for your comment on the novelty and significance of our paper. While we agree that the personalization concept has been proposed by prior research, we highlight the unintended and cost-free improvement in fairness achieved through the personalization technique within the FL paradigm. Consequently, it becomes natural to explore the fairness impact using established personalized FL algorithms, such as clustered FL and MAML FL.
> Furthermore, we provide analytical evidence demonstrating this cost-free fairness improvement under specific conditions. These findings also suggest that it could lead to a more favorable trade-off between fairness and accuracy. A new personalized FL algorithm that has a more favorable trade-off is one of our ongoing works.
>
> **Major comment 2**: Thank you for your comment on the assumptions we used in our analysis. In Proposition 2, we relax the equalized label participation rate assumption; however, we acknowledge the presence of additional assumptions, as highlighted, constitutes one of the limitations, as also mentioned in the conclusion section. To substantiate our claims, we conducted experiments using synthetic datasets where these assumptions—such as equalized label participation rate, equalized gap between label distribution, and equalized group rate—are relaxed. The results of these experiments are presented in Table 1, 2, 3, and Appendix D. We also agree that a thoughtful and rigorous theoretical analysis not relying on these assumptions would be worth exploring in future studies.

---

### Official Review · Reviewer_4EZY · 2023-11-06

**Soundness:** 2 fair
**Presentation:** 3 good
**Contribution:** 2 fair
**Rating:** 5
**Confidence:** 4

**Summary:**

This paper investigate the impact of personalization in federated learning for fairness. The analysis show that under certain constraints, introducing personalization techniques could achieve a better accuracy-fairness tradeoff. Empirical and theoretical analyses on real-world adult dataset and synthetic dataset supports the authors' claims.

**Strengths:**

S1. Interesting study to show the relationship between fairness, accuracy, and personalization.

S2. Both empirical and theoretical analyses are provided.

**Weaknesses:**

W1. When motivating fairness in federated learning, the authors use 2 examples, and I am concerned about both examples. For the 1st example, I doubt federated learning is a commonly used learning paradigm for LLMs. And for the 2nd example, it seems the mentioned paper is not related to federated learning either. Is it possible to offer stronger evidence to motivate fair federated learning?

W2. The assumption that clients within the same cluster are identical is too strong. This might be a too simple case for data heterogeneity and can be seen as FL with only 2 clients in my opinion (is it better to describe the scenario in this way?). What is the rationale for this assumption? And what is the main theoretical difficulty of assuming a more complex case?

W3. What are the x-axis and y-axis in Figures 1, 2, and 3? Is each bar a bin w.r.t. fractions? Is y-axis the normalized count of states within that bin?

W4. Why can we assume that $\mu_b^0 \leq \mu_a^0 \leq \mu_b^1 \leq \mu_a^1$?

W5. While I appreciate the interesting analysis in this paper, it would be better to have a way to summarize the key findings, e.g., a table or itemized list to show under which condition personalization is recommended to obtain fairness for free.

W6. Showing such analyses for statistical parity is great. Is there any technical limitation for analyzing equal opportunity? How about Rawls' fairness that maximize the worst-off group accuracy?

**Questions:**

Please see weaknesses.

---

> ### Author Response · Authors · 2023-11-21
> **Thank you for your valuable suggestions and comments. We address the questions in the comments below.**
>
> Thank you for your valuable comments and suggestions.
>
> **Major comment 1**: Thank you for your careful review of the motivation of fairness in FL. Training a fair ML model in decision-making scenarios is very important, and it has received lots of attention in centralized ML. Fair models can help decision-makers mitigate (potential) discrimination across different demographic groups (e.g., race, gender) in many scenarios such as hiring and loan application. However, challenges arise in decentralized ML, where each client's data is kept confidential locally. The FL paradigm has the advantage of collaborative training among multiple clients. Moreover, data heterogeneity among clients is a common characteristic in FL, which may lead to discrimination against underprivileged groups defined by protected attributes. Hence, learning a fair FL while maintaining the advantage of collaborative training and privacy-preserving becomes crucial and necessary.
>
> **Major comment 2**: Thank you for your careful reading. You are correct that by assuming clients within the same cluster are identical, the FL framework could be viewed as having only two clients. We introduce this assumption for simplicity to derive closed-form solutions for both clustered FL and FedAvg algorithms, and to investigate the different fairness impacts under these solutions. We agree that this assumption is strong, and if relaxed, the FedAvg solutions would not only be influenced between two clusters but also by variations among clients within each cluster.
>
> **Major comment 3**: Thank you for your careful reading. You are correct that the x-axis is the fraction value shown in plot a), and the y-axis is the normalized count of states.
>
> **Major comment 4**: Thank you for your comment on the metric of fairness we used throughout this paper. You are correct that this is one of the limitations as we have also mentioned in the conclusion section. We will add more experiment results using diverse fairness metrics (e.g., equal opportunity, Rawl's fairness, etc.) in our revised draft.
>
> **Major comment 5**: Thank you for your comment on the assumption used in our analysis. For simplicity, we consider one-dimensional feature values for different labels and groups. To simulate the distinction in feature values between label 1 (qualified) and label 0 (unqualified) samples, we consider $\mu^0_g \leq \mu^1_g$. Furthermore, to mirror the variation in feature values between the underprivileged and privileged groups, we consider $\mu^y_b \leq \mu^y_a$. Consequently, there are two possible outcomes: 1): $\mu^0_b \leq \mu^1_b \leq \mu^0_a \leq \mu^1_a$ and 2): $\mu^0_b \leq \mu^0_a \leq \mu^1_b \leq \mu^1_a$. For outcome 1), it implies that the average feature value of qualified samples in the underprivileged group is even lower than that of unqualified samples in the privileged group, which is an uncommon scenario. Therefore, we opt for the case $\mu^0_b \leq \mu^0_a \leq \mu^1_b \leq \mu^1_a$.
>
> **Minor comment 1**: Thank you for your careful reading. We will summarize our key findings and contributions clearly in the revised draft.

---

> > ### Comment · Reviewer_4EZY · 2023-11-23
> > **Thank you for the response**
> >
> > Thank you for providing the response. Most of my concerns have been addressed, but I will keep my score unchanged for the following two reasons:
> > - For major comment 1, I understand that FL is important for decentralized ML and can help with privacy preservation and data heterogeneity, but my concern is that modern foundation model/LLM and YouTube video recommender system is not trained with FL, so both examples may not be suitable to motivate a research in federated learning.
> > - For major comment 2, as agreed by the authors, such assumption might be too strong. Unless a reasonable justification how this can hold in the real-world can be provided, it is hard to understand the significance of the theoretical analysis.

---

### Official Review · Reviewer_eX39 · 2023-11-07

**Soundness:** 3 good
**Presentation:** 2 fair
**Contribution:** 2 fair
**Rating:** 5
**Confidence:** 4

**Summary:**

This paper studied the cost-free fairness brought by personalized federated learning. With clustering clients into groups, the authors showed that personalization in FL could lead to fairness in the FL paradigm. Finally, experiments were conducted on several datasets to investigate the impact of personalization techniques.

**Strengths:**

S1. This study demonstrated that personalization in FL can promote fairness, even without the use of dedicated fair FL algorithms. This represents a novel and intriguing discovery.

S2. This study employed numerical experiments to support its findings and conclusions.

S3. The figures included in this study served as illustrative representations of the findings.

**Weaknesses:**

W1. The paper considered only one effectiveness metric for fairness, ASPD, and hence it may not be entirely convincing, as it provides a limited scope for evaluating fairness. In particular, the fairness in FL is not a new problem.

W2. It is worth noting that while the study claims that the samples are drawn independently and identically distributed (IID) in the conclusion section, this is not explicitly stated in the experiment section. Moreover, an IID setting is less practical than real-world FL application.

W3. In the third experiment, only one normalized sample frequency is considered, i.e., Fig.3 (b) may not describe the situation well. For the fraction is comparable for label 0, but differs for label 1.

W4. The comparison with fair FL may not be entirely convincing due to the inadequate experiment results and the inclusion of only one baseline and its variant. Further experiments and additional baselines could strengthen the credibility of the comparison.

W5. The introduction in Section 3 regarding FL algorithms may contain some inaccuracies. For instance, the clustered FL algorithm (Ghosh et al.) is described as clustering based on model similarity, whereas in reality, it clusters based on model performance.

**Questions:**

Beyond the above weak points, there are also additional questions:

Q1. This paper does not conduct experiments under a non-iid setting, which may affect the overall persuasiveness of the results. What about experimental evaluation under Non-IID setting? Does your solution still work well?

Q2. Following the question above, do we really need personalization under iid setting? Please provide more justifications, references, and real-world applications to demonstrate the motivation.

Q3. In the experiment, each client is provided with 1000 training samples and 2000 testing samples, why is the size of testing samples twice of training samples?

---

> ### Author Response · Authors · 2023-11-21
> **Thank you for your valuable suggestions and comments. We address the questions in the comments below.**
>
> Thank you for your valuable comments and suggestions.
>
> **Major comment 1**: Thank you for your comment on the metric of fairness we used throughout this paper. You are correct that this is one of the limitations as we have also mentioned in the conclusion section. We will add more experiment results using diverse fairness metrics in our revised draft. In addition, you are also correct that fairness in FL is not a new problem, but what we want to highlight is the unintended fairness benefits brought through the personalization technique in FL.
>
> **Major comment 2**: Thank you for your comment on the IID setting in our experiments. In our experiments using synthetic data (Table 1,2,3) and analysis, we indeed assume IID samples within each client, but the data generating distributions are different among clients. This is further evidenced in our experiments using real-world datasets (Fig. 1,2,3). We agree that the IID assumption in our analysis is one of the limitations. A thoughtful and rigorous theoretical analysis not relying on the IID assumption would be worth exploring in future studies.
>
> **Major comment 3**: Thank you for your careful review of our comparison with fair FL algorithms. You are correct that we only compare with one existing fair FL algorithm (and its variation). In the corresponding experiment, what we want to emphasize is that if the objective is to improve fairness in FL, there are fruitful fair algorithms that we could use. However, we point to the cost-free local fairness improvement achieved through the personalization technique. In other words, the incorporation of personalization techniques provides a dual benefit in improving both local fairness and accuracy.
>
> **Major comment 4**: Thank you for your comment on the need of personalization with IID setting. You are right if all clients are the same, and samples within each client are also IID, then there is no need for personalization. However, what we assumed is the samples within each client are IID, but clients are heterogeneous. In this case, incorporation of the personalization technique could be beneficial to the local accuracy for each client. For example, in [1], they consider $m$ local training datasets (clients), and each dataset is sampled from one of $k$ different data generating distribution; in [2], they consider $m$ different machines (clients) partitioned into $k$ disjoint clusters, and for each machine, it contains $n$ iid samples points; in [3], they consider $n$ users, and samples in each user $i$ are sampled from a joint distribution $p_i$ over $(\mathcal{X}_i,\mathcal{Y}_i)$, where $p_i \neq p_j$ if $i \neq j$. We will add more explanations in the revised draft to avoid potential confusion.
>
> [1] Sattler, Felix, Klaus-Robert Müller, and Wojciech Samek. "Clustered federated learning: Model-agnostic distributed multitask optimization under privacy constraints." IEEE transactions on neural networks and learning systems 32.8 (2020): 3710-3722.
>
> [2] Ghosh, Avishek, et al. "An efficient framework for clustered federated learning." Advances in Neural Information Processing Systems 33 (2020): 19586-19597.
>
> [3] Fallah, Alireza, Aryan Mokhtari, and Asuman Ozdaglar. "Personalized federated learning: A meta-learning approach." arXiv preprint arXiv:2002.07948 (2020).
>
> **Minor comment 1**: Thank you for your careful reading in the third experiment. Due to the page limit, we only include one normalized frequency plot for each experiment in the main paper, but we have a full plot for each experiment in the Appendix.
>
> **Minor comment 2**: Thank you for your careful reading. We use a larger test dataset to ensure an accurate estimation of the fairness gap w.r.t. clients' test distributions. We will also revise the inaccuracies sentences and add more clarifications and explanations in the revised draft.

---

> > ### Comment · Reviewer_eX39 · 2023-11-23
> > **Response to the author feedback**
> >
> > Thank you for the detailed response.

---

### Meta-Review · Area_Chair_YUKX · 2023-12-11

**Metareview:**

The paper studies whether the use of personalization in federated learning (FL) can enhance fairness. The reviewers brought up several issues with the paper. First, the study's focus is on a single metric for fairness evaluation--statistical parity--which is not very well-motivated in the FL setting. A natural question that might strengthen the paper is whether the findings generalize to other metrics of fairness. The reviewers also pointed out that the assumptions in the paper are too strong for realistic settings of FL. The experiments can also be more thorough with more datasets and baselines. Finally, the reviewers made some suggestions to improve the writing.

**Justification For Why Not Higher Score:**

There is no support for acceptance.

**Justification For Why Not Lower Score:**

N/A

---

### Decision · Program_Chairs · 2024-01-16

Reject